# Hydrophobic Modified Ceramic Aeration Membrane for Effective Treatment of Brine Wastewater

**DOI:** 10.3390/membranes13040443

**Published:** 2023-04-19

**Authors:** Xinqiang Xu, Hua Zhang, Jiang Jin

**Affiliations:** College of Materials Science and Engineering, Nanjing Tech University, Nanjing 211816, China; xxq15156232801@163.com

**Keywords:** high-porosity ceramic membrane, hydrophobic modification, brine wastewater, aeration evaporation

## Abstract

A novel approach to evaporate brine wastewater using a ceramic aeration membrane was proposed. A high-porosity ceramic membrane was selected as the aeration membrane and was modified with hydrophobic modifiers to avoid undesired surface wetting. The water contact angle of the ceramic aeration membrane reached 130° after hydrophobic modification. The hydrophobic ceramic aeration membrane showed excellent operational stability (up to 100 h), high salinity (25 wt.%) tolerance, and excellent regeneration performance. The evaporative rate reached 98 kg m^−2^ h^−1,^ which could be restored by ultrasonic cleaning after the membrane fouling occurred. Furthermore, this novel approach shows great promise for practical applications toward a low cost of only 66 kW·h·m^−3^.

## 1. Introduction

The rapid global industrial expansion raises living standards, but it also has a major negative environmental impact. The increase in industrial effluent poses a major threat to the environment and public health [1,2,3]. The shortage of freshwater is another serious problem. Seawater accounts for 96% of global water resources, so obtaining freshwater from seawater and industrial wastewater can lessen freshwater scarcity and minimize wastewater pollution [4,5,6]. The traditional technology for treating brine wastewater is evaporating the wastewater from evaporation ponds [7], which brings not only serious pollution to the land but also wastes many recoverable resources. Minimal liquid discharge (MLD) is a wastewater treatment strategy to eliminate brine wastewater and recover freshwater, which has become one of the major research topics in wastewater treatment in recent years [8,9,10]. Currently, two main technologies for MLD have been developed. One is membrane-based technologies, which can be represented by reverse osmosis [11,12], and the other is thermal-based technologies, represented by multi-effect distillation and evaporative crystallization [13,14]. Membrane-based technologies provide the advantages of low energy consumption and high efficiency. The majority of membrane materials used in membrane-based technologies are polymers, such as cellulose acetate and polyamide. Sana Abdelkader et al. treated brine wastewater by direct contact membrane distillation and obtained high permeate quality. Moreover, the conductivity is reduced by 99.9%, and the chemical oxygen demand is removed by more than 99.9% [15]. Recently, Wang et al. used covalent organic framework (COF) membranes to treat seawater brine. This process achieved ultrafast desalination (267 kg m^−2^ h^−1^). The COF membrane exhibits excellent sodium chloride rejection (99.91%) [16]. Aines et al. noted that conventional reverse osmosis technology could achieve a salinity tolerance of 8.5% with only 10% water recovery [17]. However, membrane-based technologies have the issues of low tolerance to high salinity and low fouling resistance. Sodium chloride crystallizes at high concentrations and deposits in the membrane pores if the sodium chloride solution wets the membrane. The increased salinity in brine wastewater brings serious membrane fouling, leading to a significant decrease in desalination efficiency. Thermal-based technologies that rely on evaporation and condensation exhibit perfect high salinity tolerance and freshwater recovery efficiency (up to 99%) [7,18]. Iraj Ghofrani et al. reported that a bubble-column humidification-dehumidification is coupled with a multiple-effect distillation/vapor compression system in a novel way to overcome the high water cost of conventional zero liquid discharge systems [19]. However, thermal-based technologies need energy to heat the brine wastewater. In some plants, the thermal energy for heating the brine wastewater can be provided by industrial waste heat; however, it requires the temperature of industrial waste heat to reach several hundred degrees Celsius, which largely limits the application of thermal-based technologies. In addition, the traditional industrial metal evaporators of thermal-based technologies are easily corroded, affecting the equipment’s working stability.

The evaporation area is another important factor that affects evaporation efficiency [20]. The evaporation area can be effectively increased using the aeration process. Liu et al. reported that a hollow fiber hydrophobic membrane was used for high-density micro-bubble aeration to treat brine [21]. Air bubbles brought water vapor out of the brine to produce pure water and disturb the solution. The evaporation rate of brine was accelerated due to increased evaporation area; however, more energy was consumed during the aeration process because of the high resistance of the hollow fiber hydrophobic membranes. Furthermore, in their membrane aeration process, the brine was heated to make more water vapor absorbed by bubbles, which also rely on thermal energy to some extent.

An overview of the MLD treatment technologies is provided in Table 1. The table shows that there is no single technology to achieve MLD perfectly. Due to permeability limitations, reverse osmosis can only be salinity resistant up to 7%, and thermal-based multi-effect distillation must be constructed with expensive corrosion-resistant materials. The brine concentrators showed excellent salinity tolerance; however, their very high cost is a limiting factor for applicability. Membrane distillation exhibits good permeability without heating to boiling, but there is a potential for membrane fouling. Some emerging technologies (e.g., bubble membrane crystallization) show promise in avoiding membrane fouling. However, there are still some other issues, such as high aeration resistance.

To address the lack of the above technologies for treating brine wastewater, we propose a novel approach to evaporate brine wastewater using a ceramic aeration membrane. The objective is to concentrate brine wastewater and recover freshwater. The feasibility and optimal experimental conditions of the approach are mainly discussed. The sodium chloride solution was used to simulate brine wastewater. Microscopically, the molecules in the liquid are in constant irregular motion, and a part of the molecules with high kinetic energy will leave the surface of the liquid. This is the principle of evaporation. Three main factors affect the evaporation rate: temperature, evaporation area, and gas–liquid contact time. In the process, many microbubbles are generated through membrane aeration. The bubbles can agitate the liquid and increase the evaporation area to accelerate the evaporation rate. On the other hand, industrial waste heat was introduced through the ceramic aeration membrane to increase the air temperature. Theoretically, the saturated water vapor content increases with the increase in air temperature. The evaporation rate increases with the saturated water vapor content because more water vapor can be evaporated into the same air volume. In addition, the liquid column height of the aeration tank affects the gas–liquid contact time. The higher the height of the liquid column of the aeration tank is, the longer the gas–liquid contact time is. The influence of air temperature, wind speed, and liquid column height on evaporative rate was investigated. The ceramic membrane with high porosity was used as a ceramic aeration membrane to reduce the resistance of the membrane and lower energy consumption. The ceramic aeration membrane was modified by a suitable hydrophobic agent to improve the fouling resistance of the membrane. The hydrophobicity and working stability were determined. Furthermore, the high salinity tolerance of the hydrophobic ceramic aeration membrane was evaluated.

## 2. Materials and Methods

### 2.1. Preparation of the Ceramic Aeration Membrane

A high-porosity ceramic membrane made of alumina fiber was used as the ceramic aeration membrane. The preparation of high porosity ceramic membrane was described in previous work [22,23]. The alumina fiber, binder, glass powder, and water were mixed thoroughly, and the mixture was pressed into a cylindrical green body (ϕ 25 × 3 mm), followed by sintering at 1300 °C. The sample’s porosity and mean pore size are 76.0% and 0.20 µm, respectively.

The surface of the high-porosity ceramic membrane was modified using polytetrafluoroethylene (PTFE) or polydimethylsiloxane (PDMS) to obtain surface hydrophobicity. The high-porosity ceramic membrane was cleaned in an ultrasonic cleaner (F-010S, Fuyang Technology Group Co., Ltd., Shenzhen, China, 40 KHz) with ethanol for 15 min and dried in an oven at 80 °C for 1 h. The cleaned ceramic membrane was impregnated with PTFE dispersion (10 wt.%, China National Pharmaceutical Group Co., AR. Shenzhen, China) at room temperature for 20 min. Then, the ceramic membrane was transferred to a tubular oven and heated at 250 °C for 1 h. The PDMS (20 wt.%, China National Pharmaceutical Group Co., AR. Shenzhen, China), ethyl silicate (Aladdin Reagent Co., Ltd., AR. Shanghai, China), dibutyltin dilaurate catalyst (Aladdin Reagent Co., Ltd., AR. Shanghai, China), and n-heptane (China National Pharmaceutical Group Co., AR. Shenzhen, China) were mixed thoroughly to obtain the PDMS modified solution. The cleaned ceramic membrane was impregnated with the solution at room temperature for 20 min. Then, the ceramic membrane was heated at 200 °C for 1 h, followed by raising the temperature to 400 °C and keeping it for 2 h. In particular, the heating process was carried out in an atmosphere of H_2_/N_2_ (flow ratio H_2_:N_2_ = 5:95) to prevent oxidation.

### 2.2. Evaporation Apparatus

The process of evaporation using a ceramic aeration membrane is shown in Figure 1. Red arrows indicate the airflow direction. Negative air pressure was provided by the suction of the vortex blower (YASHIBA HG-90B, 120 W, Ningbo, Zhejiang, China). The simulated industrial waste heat was supplied by a hot-air chamber (DELI DL391160, 1600 W, Shanghai, China). The flow rate was controlled by an airflow valve connected to the vortex blower and then monitored by a wind speed meter (SMART SENSOR AR866A, Shenzhen, Guangdong, China). Blue arrows indicate the liquid flow direction. The feed solution was water or sodium chloride solution (3 wt%). The liquid in the aeration tank was supplied from the feed tank, and the flow rate was controlled by a liquid flow valve. The remaining liquid was discharged to the concentrated tank through the lower outlet after the solution in the aeration tank had evaporated to a saturated concentration (25 wt%). The water vapor generated in the aeration tank is condensed in the produced water tank.

### 2.3. Characterization of the Ceramic Aeration Membrane

The microstructure and morphology of the ceramic aeration membrane were observed by scanning electron microscope (SEM, JSM-6510, JEOL Ltd., Tokyo, Japan). The loading of the hydrophobic modifier on the ceramic aeration membrane was analyzed by a thermal gravimetric analyzer (TG, NET ZSCH STA 449 F5, Free State of Bavaria, Germany). The water contact angle on the surface of the modified hydrophobic ceramic membrane was examined by a contact angle meter (WCA, KRUSS DSA-100, Hamburg, Germany).

### 2.4. Performance Tests

The evaporative rate was calculated by the following equation [21]:$$F=\frac{W_{m}}{A\cdot t}$$

F is the evaporative rate, kg m^−2^ h^−1^; Wm is the evaporative mass, kg; A is the area of the ceramic aeration membrane, m^2^; and t is the operating time, h.

The working stability was tested using sodium chloride solution (3 wt.%) as a feed liquid. The operation condition was set as a wind speed of 1.2 m/s, air temperature of 50 °C, and liquid column height of 50 mm. The evaporative rate was recorded every hour.

The aeration resistance of the ceramic aeration membrane was measured using a U-type manometer.

The ceramic aeration membranes were immersed in the aqueous solutions with pH 2–12 for a chemical stability test. After 24 h, the ceramic aeration membranes were removed and washed with ethanol. Then, the ceramic aeration membranes were dried at 80 °C for 2 h. Finally, the water contact angle and evaporative rate were tested.

The evaporation efficiency was expressed by the evaporative rate. The influence of air temperature, wind speed, and height of the liquid column on the evaporative rate was analyzed separately.

The high salinity tolerance was tested using sodium chloride solution (3 wt.%) as a feed liquid. The wind speed, air temperature, and height of the liquid column were controlled at 1.2 m/s, 50 °C, and 50 mm, respectively. The evaporation rate was tested until the evaporation rate dropped deeply. Then, the ceramic aeration membrane was ultrasonically cleaned for 10 min, and the evaporative rate was detected again.

## 3. Results and Discussion

### 3.1. Characterization of Modified Ceramic Aeration Membrane

The connected porous structure of a high-porosity ceramic membrane provides excellent permeation fluxes. The ceramic membrane is modified with PDMS and PTFE to obtain surface hydrophobicity. The microstructure and morphology of ceramic aeration membranes are shown in Figure 2. The fiber in the unmodified ceramic membrane shows a smooth surface (Figure 2a,b). In contrast, numerous small spherical particles are attached to the fiber surface in the PDMS-ceramic aeration membrane (Figure 2c,d). A layer is covered on the surface of the fiber in the PTFE-ceramic aeration membrane (Figure 2e). The result of EDS mapping showed the presence of the F element on the fiber of the ceramic membrane, further proving the success of the PTFE modification (Figure 2f). As seen in Figure 3, the water contact angle of the unmodified ceramic membrane is 0° (Figure 3a), while the ceramic aeration membranes modified by PDMS and PTFE show water contact angles of 139.9° and 130.5°, respectively (Figure 3b,c). The modified ceramic aeration membranes exhibited outstanding hydrophobicity.

The loading of the hydrophobic modifier on the ceramic aeration membrane is analyzed by thermal gravimetric analysis. From Figure 4, the PDMS-ceramic aeration membrane significantly lost weight between 340 and 420 °C with an overall weight loss of 3.32%. The PTFE-ceramic aeration membrane shows an obvious weight loss of 2.88% between 520–590 °C. The overall weight loss is mainly caused by the pyrolysis of the hydrophobic modifiers. The loading of PDMS and PTFE is estimated to be around 3% based on weight loss. The aeration resistance of the ceramic aeration membrane increases with the load of the modifiers. It needs more energy to overcome the aeration resistance during the process. As shown in Table 1, the air resistance in the unmodified ceramic membrane is 1200 Pa, and the air resistance for PDMS-modified and PTFE-modified ceramic aeration membranes increases to 1800 Pa and 1860 Pa, respectively. However, the aeration resistance increases rapidly to 3740 Pa once the unmodified ceramic membrane is in contact with the feed liquid (Table 2), while the aeration resistance of PDMS-modified and PTFE-modified ceramic aeration membranes remains the same when they are in contact with the feed liquid. This demonstrates that the hydrophobic modification for ceramic aeration membrane is necessary to lower the aeration resistance and reduce energy consumption.

### 3.2. Effect of Hydrophobic Modifier on Working Stability

The evaporative rate using PDMS-ceramic aeration membrane reaches 43 kg m^−2^ h^−1^ and remains stable during the continuous operation of 100 h (Figure 5a). The membrane surface is continuously purged during aeration, enhancing the ability to avoid membrane contamination. However, the evaporative rate using a PTFE-ceramic aeration membrane shows a rapid decrease after 20 h. The surface morphology of the failed PTFE-ceramic aeration membrane is observed by SEM (Figure 5b,c). It can be seen that the crystalline grains appear on the surface, and part of the pores is blocked by these crystalline grains (Figure 5c), which causes a rapid decrease in the evaporative rate. The water contact angle on the failed PTFE-ceramic aeration membrane is measured again and decreases to 0°, implying that the PTFE modifier peels off from the membrane due to the non-stick characteristics of PTFE [24,25]. The binding between PTFE and the ceramic membrane is not strong, and the PTFE modifier is easy to peel off, which leads to the failure of the PTFE-ceramic aeration membrane. The failed PTFE-ceramic aeration membrane is wetted by sodium chloride solution during aeration, and sodium chloride crystallizes in the membrane pores, causing membrane fouling. Thus, the PDMS-ceramic aeration membrane is used in the subsequent tests.

The chemical stability of PDMS-ceramic aeration membranes is studied by the aqueous solutions with pH 2–12. The operating conditions are set as an air temperature of 50 °C, height of the liquid column of 50 mm, and wind speed of 1.2 m/s. The water contact angle of the PDMS-ceramic aeration membrane is maintained at 138.1° (±1°) after immersion of 24 h, and the evaporative rate remains at 43 (±1) kg m^−2^ h^−1^ (Figure 5d). It is obvious that the PDMS-ceramic aeration membrane has good chemical and working stability. In summary, the PDMS-ceramic aeration membrane can meet the demands of aeration evaporation.

### 3.3. Effect of Aeration Conditions on Evaporative Rate

In this work, many micron-sized bubbles are produced to improve the evaporation area. The air volume passed through the same membrane area directly affects the number of bubbles. The larger the air volume is, the more the number of bubbles is. The air volume that passes through the unit membrane area in a given time is determined by wind speed. The influence of wind speed on the evaporative rate at an air temperature of 50 °C and a liquid column height of 50 mm is shown in Figure 6a. More bubbles are produced as wind speed increases and the evaporation area is significantly enlarged. The upper limit of wind speed in this study is 1.2 m/s because the gas–liquid contact time will be reduced significantly if the wind speed is too high.

The gas–liquid contact time increases with the increase in the height of the liquid column. Theoretically, the increase in the height of a liquid column of the aeration tank benefits evaporation. However, the aeration pressure also increases with the increase in the height of the liquid column of the aeration tank. The larger the aeration pressure is, the less the air volume is. The influence of the height of the liquid column on the evaporative rate at an air temperature of 50 °C and a wind speed of 1 m/s is shown in Figure 6b. The aeration pressure increases with the increase in the height of the liquid column, and the evaporative rate decreases significantly. When the liquid column height reaches 140 mm, the aeration pressure increases to 3220 Pa, and the evaporative rate decreases to 39 kg m^−2^ h^−1^. As the height of the liquid column increases further, the aeration pressure and the evaporative rate remain constant.

The air temperature is another important factor that affects the evaporative rate. Theoretically, the evaporative rate increases with the increase in saturated water vapor content because more water vapor can be evaporated into the same volume of air. The saturated water vapor content increases with the increase in air temperature. Therefore, raising the air temperature can take away more water vapor and achieves a higher evaporative rate. The effect of the air temperature on the evaporative rate at a liquid column height of 50 mm is illustrated in Figure 6c. It shows an upward trend as air temperature increases. The evaporative rate increases from 20 kg m^−2^ h^−1^ to 98 kg m^−2^ h^−1^ as the air temperature elevates from 30 to 90 °C at a wind speed of 1.2 m/s. On the other hand, more air volume is brought in as wind speed increases. The influence of the air temperature on the evaporative rate becomes more pronounced at higher wind speeds. The upper limit of air temperature in this study is merely 90 °C, which can be supplied by industrial waste heat generated by numerous plants. Compared with thermal-based technologies, brine wastewater absorbs heat energy directly from industrial waste heat, and the energy consumption for heating can be eliminated in this approach.

### 3.4. High Salinity Tolerance of Ceramic Aeration Membrane

The high salinity tolerance is evaluated using 3 wt.% sodium chloride solution for aeration evaporation, and the evaporative rate is determined until it drops deeply. The porous structure of PDMS-modified ceramic membrane at different evaporation stages is shown in Figure 7. It can be seen that PDMS attaches to the surface of the ceramic aeration membrane at the start of the operation (Figure 7a). Meanwhile, the sparse, porous structure of the ceramic membrane is retained, which is very conducive to creating bubbles by aeration. The evaporative rate is measured as 43 (±1) kg m^−2^ h^−1^. Figure 7b indicates that the surface of the membrane still exhibits an outstanding sparse and porous structure even if the solution is evaporated to near saturation (25 wt.%). The evaporative rate maintains at 43 (±1) kg m^−2^ h^−1^ at a high concentration of sodium chloride solution. The sodium chloride crystals are precipitated as sodium chloride solution tends to saturation, and the evaporative rate decreases to 0 kg m^−2^ h^−1^ rapidly. There is obvious particulate matter in the membrane pores, and most of the pores are blocked by the particulate matter (Figure 7c). The hydrophobicity of the failed membrane is subsequently examined. It is noteworthy that although the PDMS-ceramic aeration membrane shows an extremely rapid decrease in evaporative rate, the failed PDMS-ceramic aeration membrane still maintains excellent hydrophobicity. The above PDMS-ceramic aeration membrane is ultrasonically cleaned for 10 min. Then, it is placed in an oven and heated at 80 °C for 1 h. The membrane pores are re-exposed (Figure 7d), and the evaporative rate is restored to 43 (±1) kg m^−2^ h^−1^. Compared with the membrane-based technologies for MLD, PDMS-ceramic aeration membrane shows perfect high salinity (25 wt.%) tolerance due to the continuous purge of the membrane surface during aeration, which makes it more promising for application. In addition, the PDMS modifier shows a perfect combination with the ceramic aeration membrane because of the excellent stability of hydrophobic SiCO nanoparticles in PDMS and their strong adhesion to alumina [26].

### 3.5. Economic Performance

In this approach, the energy for aeration evaporation relies on the air volume and temperature. The low-temperature industrial waste heat is used as a gas source for aeration, and the enormous energy for heating the brine wastewater is eliminated. The majority of the energy consumption is electrical to provide air volume, which is provided by the vortex blower. The maximum evaporative rate of 98 kg m^−2^ h^−1^ can be achieved at a wind speed, air temperature, and height of the liquid column of 1.2 m/s, 90 °C, and 50 mm, respectively. The required air pressure and volumes are 3000 Pa and 4320 m^3^, respectively. Treating one cubic meter of industrial brine wastewater requires a vortex blower with a power of 6 kW to operate for up to ten hours. It is estimated that 60–66 kW·h·m^−3^ of electricity is required.

## 4. Conclusions

In this work, industrial brine wastewater is evaporated and concentrated to saturation concentration using a PDMS-ceramic aeration membrane. The maximum evaporative rate of 98 kg m^−2^ h^−1^ can be attained at a wind speed of 1.2 m/s, air temperatures of 90 °C, and a height of the liquid column of 50 mm.

The PDMS-ceramic aeration membrane has excellent hydrophobicity and chemical stability. In addition, PDMS-ceramic aeration membranes can operate continuously for more than 100 h. These results demonstrate that the PDMS-ceramic aeration membrane is more suitable for treating brine wastewater. On the other hand, although PTFE can also be used for hydrophobic modification of ceramic membranes, it is not applicable to this approach because PTFE will be peeled off during aeration.

The membrane fouling resistance is improved even in high salinity (25 wt.%) because the membrane surface was continuously purged during aeration. The industrial waste heat is introduced through the ceramic aeration membrane to heat brine wastewater and increase the evaporation area. The energy consumption for heating is eliminated. This work presents a new feasible example of treating brine wastewater.

Compared to the commercially available technologies, this technology compensates for the low salinity tolerance in membrane-based technologies and the high thermal energy demand in thermal-based technologies. This technology achieves efficient treatment of brine wastewater at high salinity. At the same time, the problem of membrane fouling at high salinity is alleviated. Currently, this technology only achieves a concentration of brine (evaporation of brine to saturation). The crystallization of brine needs to be studied further.

## Figures and Tables

**Figure 1 membranes-13-00443-f001:**
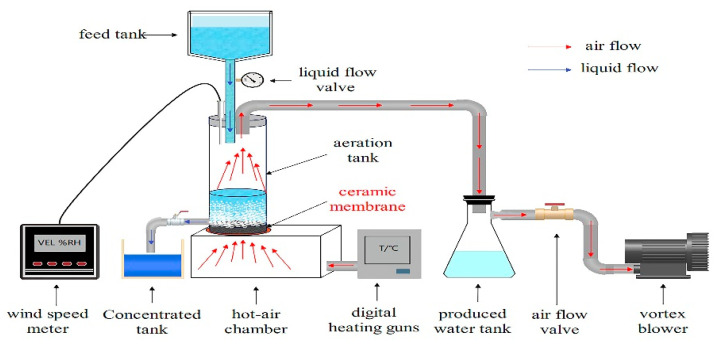
Schematic diagram of evaporation using a ceramic aeration membrane.

**Figure 2 membranes-13-00443-f002:**
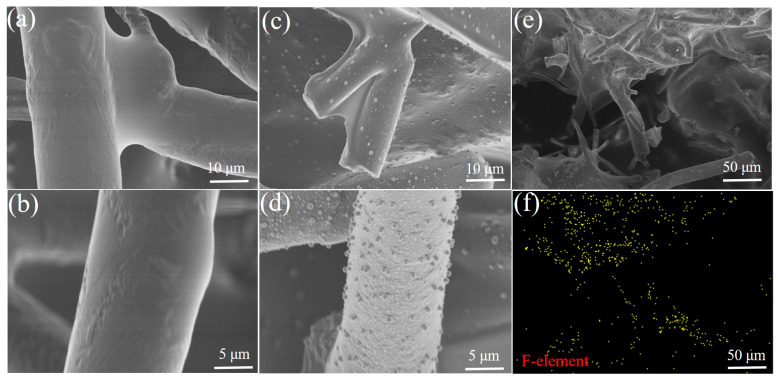
SEM and EDS images of ceramic membrane: (**a**,**b**) SEM of unmodified ceramic membrane; (**c**,**d**) SEM of PDMS-ceramic membrane; (**e**) SEM of PTFE-ceramic membrane; and (**f**) EDS mapping of PTFE-ceramic membrane.

**Figure 3 membranes-13-00443-f003:**
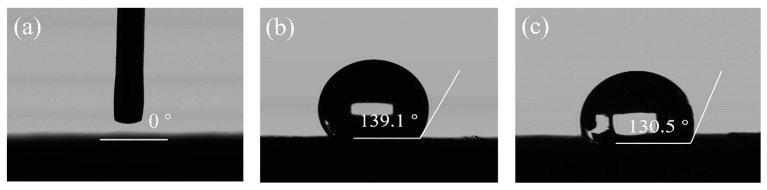
Water contact angle of ceramic membranes: (**a**) unmodified ceramic membrane; (**b**) PDMS-ceramic membrane; and (**c**) PTFE-ceramic membrane.

**Figure 4 membranes-13-00443-f004:**
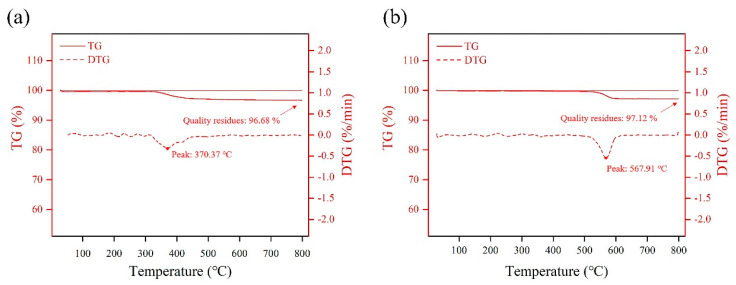
DTG and TG curves of ceramic aeration membranes: (**a**) PDMS-ceramic membrane and (**b**) PTFE-ceramic membrane.

**Figure 5 membranes-13-00443-f005:**
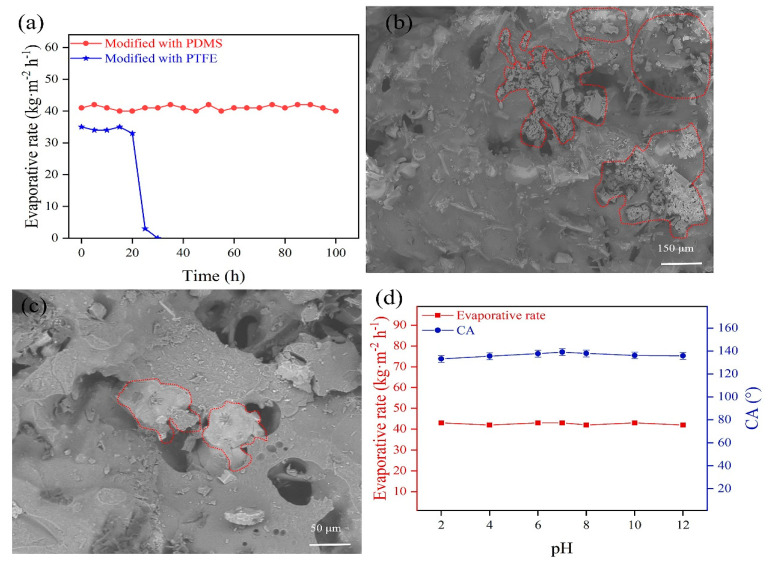
(**a**) Working stability of ceramic aeration membranes; (**b**,**c**) Membrane surface of failure of PTFE-ceramic membrane; (**d**) Chemical stability of PDMS-ceramic membrane.

**Figure 6 membranes-13-00443-f006:**
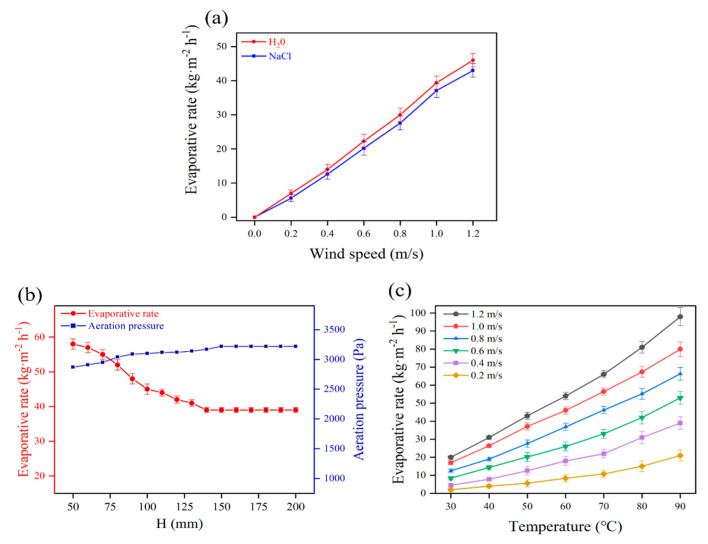
Impact of different conditions on evaporative rate: (**a**) wind speed; (**b**) height of the liquid column; and (**c**) air temperature.

**Figure 7 membranes-13-00443-f007:**
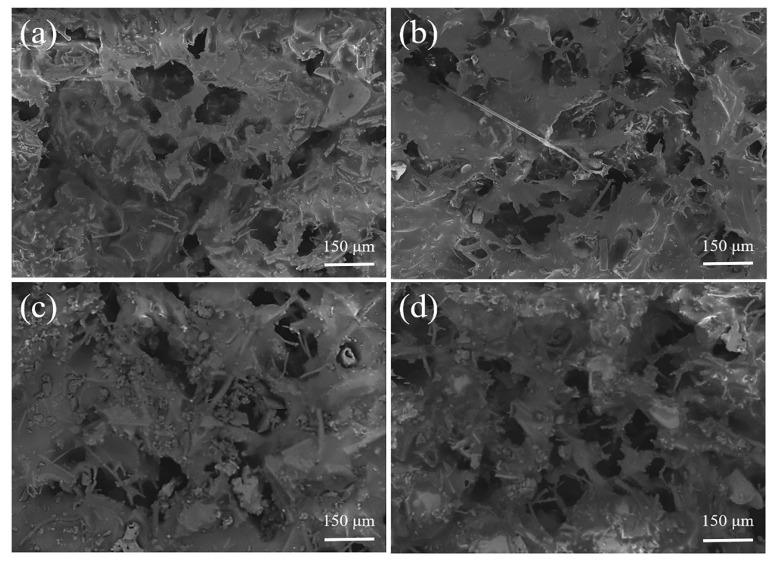
Porous structure of PDMS-ceramic aeration membrane at different evaporation stages: (**a**) original PDMS-ceramic aeration membrane; (**b**) the solution was evaporated to a nearly saturated state; (**c**) the evaporative rate decreases to 0 kg m^−2^ h^−1^; and (**d**) after recovery treatment.

**Table 1 membranes-13-00443-t001:** An overview of Minimal liquid discharge treatment technologies.

Technology	Advantages	Challenges	Salinity Tolerance (%)	Water Recovery (%)	References
Reverse osmosis	High rejection of many contaminants Less energy consumption	Needs pretreatment processes	7%	10%	[11,12]
Brine concentrators	Established technology for high salinity	Metal evaporators are easily corroded Need to heat to boiling.	30%	Up to 99%	[13,14]
Membrane distillation	No feed pressure requirements No need to heat to boiling.	Potential of membrane fouling	30%	Up to 90%	[15]
Multi-effect distillation	Produce high-quality freshwater	Needs pretreatment processes Metal evaporators are easily corroded	18%	Up to 85%	[19]
Bubble membrane crystallization	Avoid membrane fouling	High aeration resistance Need to heat to boiling.	30%	Up to 90%	[21]

**Table 2 membranes-13-00443-t002:** Resistance variation of ceramic aeration membranes.

Ceramic Aeration Membrane	Resistance (Pa)
unmodified unmodified	1200 (gas permeation resistance) 3740 (aeration resistance)
PDMS-modified	1800 (gas permeation resistance, aeration resistance)
PTFE-modified	1860 (gas permeation resistance, aeration resistance)

## Data Availability

All data that support the findings of this study are included within the article.

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
