# Peer review of "Hydrophobic Modified Ceramic Aeration Membrane for Effective Treatment of Brine Wastewater"

_membranes, 2023, doi:10.3390/membranes13040443_

Round 1
Reviewer 1 Report
1. Add your research information of this article at the end of introduction.
2. line 106: How do you measure or calculate the evaporation area?
3. Line 133, A layer is covered on the surface of fiber in PTFE-ceramic aeration membrane ((Fig. 2(e) and 2(f)): However, it is not clear at SEM pictures.
4. Line 252, perfect high salinity (25 wt.%) tolerance: Have you ever tested the ceramic membrane over 25 wt% salinity?
5. Line 265: Explain how to calculate 60-66 kW.h.m3 of electricity required to treat one cubic meters of industrial saline wastewater.
Author Response
Please see the attachment。

Reviewer 2 Report
This manuscript presents a hydrophobic modified ceramic aeration membrane for the effective treatment of brine wastewater. The PDMS-modified hydrophobic ceramic membranes showed a long-term operation stability and high salinity tolerance. Still, there are some issues needed to be clarified further:
1. It is not clear about the objective of this paper, the authors should provide some statement at the end of the introduction about the purpose of this paper and how to achieve this purpose to improve the readability of the manuscript.
2. Please change the title of section 2 ‘Materials’ into ‘Materials and Methods’ (page 2). Also, the ultrasonic cleaning (cleaning frequency and model of the ultrasonic device) of the membrane should be mentioned.
3. The resolution of Fig.1 on page 3 should be improved.
4. Please give more details about Fig. 6. What’s the wind speed in Fig.6 (b) and what’s the column height in Fig. 6(c)?
5. On page 9, how is the electricity consumption calculated? Please provide more information about the calculation.
6. Some grammar issues should be addressed.
Reviewer 3 Report
The manuscript provides an interesting discussion on high porosity ceramic membrane modified with PTFE or PDMS to obtain the surface hydrophobicity for treatment of brine wastewater. The so-modified hydrophobic ceramic aeration membrane reportedly showed good operational stability with high salinity tolerance, high evaporative rate, good regeneration performance and relatively low energy consumption. The topic is worthy of scientific investigation and the manuscript may be suitable for consideration for publication. There are, however, some points to address and that will require revision to further strengthen the manuscript.
1)
It is crucial to state what is the objective and purpose of the study. If it is to treat wastewater brine, then it is necessary to clarify (a) what level of treatment is required – in terms of water recovery or components to harvest, (b) how is ‘wastewater brine’ defined here – what are the water quality characteristics etc, and (c) provide an overall schematic to show the big picture of how this technology may fit in brine treatment or within the field of sustainability.
2)
Provide a section on literature review – if this study focuses on Minimal-Liquid-Discharge (MLD) technologies, then provide a discussion in table form to compare the various technologies and provide details such as technology principles, advantages and disadvantages, the type and quantity of the products to be recovered, Technology Readiness Level (TRL), etc…, of the different MLD technologies.
3)
Page 3, Figure 1.
Provide a detailed water analysis of all the components: feedwater tank, concentrate tank, aeration tank, produced water tank.
Note that wastewater brine does not only contain NaCl, but also other components such as Ca, Mg, CO3, SO4, organics, etc.
4)
List and explain the effects of various study parameters. What is meant by the height of the liquid column – is this the height in the feed tank or level in the aeration tank, and why is it relevant? Also, why is wind speed considered here. The study should also consider and explain other parameters – for example: rate of air supply to the aeration tank, area of the membrane, evaporation area, conductivity or total dissolved solids (TDS) of the aeration tank and concentrate tank, etc.
5)
Conduct and discuss the study using real wastewater brine as feedwater. The effect due to hydrophobicity can be severe in the presence of foulants (especially organics).
6)
Page 9, Section 3.5 Economic Performance.
The “as-is” paragraph is (too) brief. More details would need to be included – e.g. how is the estimate of 60-66 kWh/m3 derived.? What are the other economic factors – e.g. OPEX and CAPEX?
Is there any pilot or large-scale plants that are currently already using the technology?
7)
General comment. It is unavoidable to use abbreviations – e.g. MLD, COF, PTFE, PDMS, etc…For ease of reference, the manuscript may provide a section to list all abbreviations and symbols.
Also, it would be good to carry out proof reading of the manuscript, some expression e.g. (Abstract) “A high porosity ceramic membrane was modified with hydrophobic modifiers and it used as the aeration membrane to evaporate brine wastewater…” may sound awkward and may be revised to improve linguistically.
8)
Page 9. Section 4 Conclusions.
The manuscript has provided a discussion on high porosity ceramic membrane modified with PTFE or PDMS to obtain the surface hydrophobicity for treatment of brine wastewater. The conclusions can be further strengthened to link up the various points discussed and present a clear roadmap for the development of this technology, and how does it compare with commercially available technologies, and what are the areas for future R&D.
